# Beyond Exploration–Exploitation: An Identification-Aware Bayesian Optimization Method under Noisy Evaluations

## Abstract

In this study, we investigate black-box optimization problems with heteroscedastic noise, a setting commonly encountered in hyperparameter tuning for machine learning models. Bayesian optimization (BO) is a popular framework for such problems, with prior work primarily focusing on designing acquisition functions or surrogate models to balance exploration and exploitation. However, a critical yet underexplored issue is the *identification problem*: BO algorithms often locate promising solutions but fail to reliably identify and return them to users. We take the first step toward addressing this challenge. We formally define the *identification error* within a standard BO framework and derive a myopic acquisition function that directly minimizes this error. A surprising theoretical result shows that the acquisition function for minimizing identification error is equivalent to the difference between two widely used criteria: the knowledge gradient (KG) and expected improvement (EI). Building on this insight, we propose a novel acquisition function, *Identification-Error Aware Acquisition* (IDEA), and establish its asymptotic no-regret property. The effectiveness of IDEA is demonstrated on benchmark test functions.

## 1 Introduction

We study noisy black-box optimization problems, where the objective function is expensive to evaluate, has no closed-form expression, and is subject to stochastic noise. Such problems are ubiquitous in machine learning and related fields. A representative example is hyperparameter tuning, where randomness arises from sources such as parameter initialization, data shuffling, and mini-batch sampling. Bayesian optimization (BO) has emerged as a standard framework for solving these problems, precisely because it is designed to handle costly, gradient-free, and noisy evaluations. BO is a sequential sampling method and it proceeds iteratively in two steps Frazier (2018): it first builds a surrogate model that approximates the objective function based on observations from the function, and then selects new candidate solutions by optimizing an acquisition function that measures the quality of solutions. Repeating these steps allows BO to refine the surrogate and progressively guide the search toward high-quality solutions under a limited evaluation budget.

The core of BO algorithms lies in balancing *exploration* and *exploitation*, that is, trading off global and local search efforts. Consequently, extensive research has focused on designing surrogate models or acquisition functions to enhance BO's ability to discover promising solutions within computational budgets Astudillo & Frazier (2020); Wang & Li (2024). Yet an interesting question remains: even if the algorithm efficiently locates promising candidates, can it reliably identify and return them to the user? This issue is critical because, in practice, only the final solution recommended by the algorithm is implemented. Under noisy conditions, however, objective values are estimated from noisy observations, and the returned solution typically carries estimation error. As shown in our experiments, such error in identifying promising solutions can substantially degrade the reliability of BO. The issue of BO in identifying promising solutions is first proposed in Jalali et al. (2017), where the authors tested several BO methods with different acquisition functions under different pattern of heteroscedastic noise. The authors found that the issue of identifying promising solutions is a prevalent weakness of BO algorithms, especially under heavy noise. Jalali et al. (2017) termed

this issue as the *identification problem*, and they pointed out that this problem can be addressed by incorporating BO with an efficient allocation of computational resources.

Although the identification problem has been recognized, to the best of our knowledge, it has not yet been formally analyzed or addressed in literature. This work takes a first step toward filling this gap within a standard BO framework. Instead of incorporating an allocation procedure as suggested in Jalali et al. (2017), we view computational resource allocation as part of the sampling decision itself. This perspective motivates a general framework for addressing the identification problem through acquisition functions that simultaneously balance exploration, exploitation, and identification.

To this end, we first formally define the identification error in BO. Based on the definition, we derive a myopic sampling rule and uncover a surprising result: the sampling rule for minimizing identification error is equivalent to maximize the difference between two widely used acquisition function: the knowledge gradient (KG) and expected improvement (EI). This formulation is both easy to implement and amenable to theoretical analysis. Building on this insight, we propose a novel acquisition function that jointly balances exploration, exploitation, and identification, and finally we establish its convergence rate.

## 1.1 RELATED WORK

Identifying the best solution under noisy evaluations has long been the focus of the *best-arm identi-fication (BAI)* and *ranking-and-selection (R&S)* literature (Chen et al., 2000; Kim & Nelson, 2006). Given a set of candidate design, these studies investigate how to find the optimal design by allocating computational resources to these candidates. However, a key distinction from BO is that in BO the candidate set is not fixed but is sequentially expanded as the algorithm explores the search space. This distinction makes existing BAI or R&S methods not applicable because they cannot deal with the identification problem in the case that the set of candidates is expanding.

Within the BO literature, several studies have touched upon the problem of returning a reliable final solution under noise. One line of work investigates different recommendation strategies Shahriari et al. (2015), such as returning the incumbent (the best observed point) versus the posterior best (the point with the lowest posterior mean). While these heuristics can improve practical performance, their effectiveness depends heavily on the search trajectory and lacks formal guarantees on identi-fication error (Frazier, 2018). Another line of work focuses on *acquisition functions* that implicitly address noise, such as augmented EI (Huang et al., 2006) and knowledge gradient (KG) (Frazier et al., 2008; 2009). These methods can indeed improve the algorithm performance under noisy eval-uations but are not explicitly designed from the perspective of addressing the identification problem. Moreover, as reported in Jalali et al. (2017), the acquisition functions like AEI and KG also face identification problems.

Some studies have developed computational resource allocation strategies in the BO framework. Quan et al. (2013) combined EI with explicit budget allocation, while Liu et al. (2014) extended this idea to an adaptive sequential framework (eTSSO). Jalali et al. (2017) evaluated both methods and showed that they address the identification problem only partially. Similarly, Dai et al. (2023) address this issue by intelligently allocating computational resources. A key limitation, however, is that these approaches rely on manually specified allocation rules, which restricts their flexibility and scalability.

## 2 PRELIMINARIES

### 2.1 PROBLEM FORMULATION

This study considers the following stochastic black-box optimization problem:

$$\min_{\boldsymbol{x} \in \mathcal{X}} \quad \mathbb{E}_{\xi}[f(\boldsymbol{x}, \xi)], \tag{1}$$

where $\boldsymbol{x} = (x_1, \ldots, x_d) \in \mathcal{X} \subset \mathbb{R}^d$ is a $d$-dimensional decision vector, and $\mathcal{X}$ denotes a predeter-mined continuous search space. No explicit constraints are imposed.

The function $f(\boldsymbol{x}, \xi)$ represents a performance metric under decision $\boldsymbol{x}$ and a set $\xi$ of random vari-ables that introduce stochasticity into the function. As $\xi$ varies, the value of $f$ fluctuates, resulting

in stochastic (noisy) observations. Let $\epsilon_{\boldsymbol{x}}$ denote the observation noise at $\boldsymbol{x}$; we assume it follows a Gaussian distribution:

$$\epsilon_{\boldsymbol{x}} \sim \mathcal{N}(0, \tau^2(\boldsymbol{x})), \tag{2}$$

where $\tau^2(\boldsymbol{x})$ is the input-dependent noise variance.

We consider the black-box setting, where neither the analytical form of $f$ nor its gradient is available. Moreover, each function evaluation is computationally expensive. The total number of allowed evaluations is limited to $N$. The goal is to identify a solution $\boldsymbol{x} \in \mathcal{X}$ that minimizes $\mathbb{E}_\xi[f(\boldsymbol{x}, \xi)]$ within a budget of $N$ noisy observations. For brevity, hereafter we use the function $y(\boldsymbol{x})$ to denote $\mathbb{E}_\xi[f(\boldsymbol{x}, \xi)]$.

Hyperparameter tuning is a typical black-box optimization problem, where $\boldsymbol{x}$ denotes continuous hyperparameters (e.g., learning rate, regularization strength) and $f(\boldsymbol{x}, \xi)$ is the validation loss under randomness $\xi$ (e.g., initialization, data shuffling). The function is noisy, and each evaluation requires training from scratch, making it computationally expensive.

## 2.2 BAYESIAN OPTIMIZATION

Bayesian optimization is widely used in solving black-box optimization problems. At each iteration, it fits a probabilistic surrogate model and selects a candidate sample point (i.e., a solution) by optimizing an acquisition function. The candidate is then evaluated, the dataset is augmented, and the surrogate is updated. This loop continues until a pre-specified budget is exhausted or a stopping criterion is met.

### 2.2.1 GAUSSIAN PROCESS

Gaussian process (GP) is commonly used in BO to construct surrogate models for the objective function. A GP $f_{GP}$ is fully specified by a the mean function $\mu : \mathbb{R}^d \mapsto \mathbb{R}$, defined by $\mu(\boldsymbol{x}) = \mathbb{E}[f_{GP}(\boldsymbol{x})]$, and the covariance function $k : \mathbb{R}^d \times \mathbb{R}^d \mapsto \mathbb{R}$, defined by $k(\boldsymbol{x}, \boldsymbol{x}') = \mathbb{E}[(f_{GP}(\boldsymbol{x}) - \mu(\boldsymbol{x}))(f_{GP}(\boldsymbol{x}') - \mu(\boldsymbol{x}'))]$. To model the unknown function $y$, a Gaussian process prior is first assigned to the function, denoted by

$$f_{\text{GP}}(\boldsymbol{x}) \sim \mathcal{GP}(\mu_0(\boldsymbol{x}), k_0(\boldsymbol{x}, \boldsymbol{x}')). \tag{3}$$

The mean function $\mu_0$ is typically a constant function. In this study, we assume the covariance function has the form $k_0(\boldsymbol{x}, \boldsymbol{x}') = \tau^2 \rho(\|\boldsymbol{x} - \boldsymbol{x}'\|)$, where $\tau > 0$ is a constant and $\rho$ is a positive-definite kernel that decreases with distance, satisfies $\rho(0) = 1$, and $\rho(\|\boldsymbol{x} - \boldsymbol{x}'\|) \to 0$ as $\|\boldsymbol{x} - \boldsymbol{x}'\| \to \infty$.

Given a sequence of $n$ sample points $\mathcal{X}_n = (\boldsymbol{x}_1, \ldots, \boldsymbol{x}_n)$ and noisy observations at the samples $F_n = (f_1, \ldots, f_n)$, the posterior distribution of the GP can be derived in closed form. The posterior GP is:

$$f_{\text{GP}}(\boldsymbol{x}) \mid \mathcal{X}_n, F_n \sim \mathcal{GP}(\mu_n(\boldsymbol{x}), k_n(\boldsymbol{x}, \boldsymbol{x}')), \tag{4}$$

with the posterior mean and covariance are given by

$$\mu_n(\boldsymbol{x}) = \mu_0(\boldsymbol{x}) + k_0(\boldsymbol{x}, \mathcal{X}_n)[k_0(\mathcal{X}_n, \mathcal{X}_n) + \Sigma^n]^{-1}(F_n - \mu_0(\mathcal{X}_n)), \tag{5}$$

$$k_n(\boldsymbol{x}, \boldsymbol{x}') = k_0(\boldsymbol{x}, \boldsymbol{x}') - k_0(\boldsymbol{x}, \mathcal{X}_n)[k_0(\mathcal{X}_n, \mathcal{X}_n) + \Sigma^n]^{-1} k_0(\mathcal{X}_n, \boldsymbol{x}'), \tag{6}$$

where $k_0(\mathcal{X}_n, \mathcal{X}_n) = [k_0(\boldsymbol{x}_i, \boldsymbol{x}_j)]_{1 \le i,j \le n} \in \mathbb{R}^{n \times n}$, $k_0(\boldsymbol{x}, \mathcal{X}_n) = (k_0(\boldsymbol{x}, \boldsymbol{x}_i), ..., k_0(\boldsymbol{x}, \boldsymbol{x}_n)) \in \mathbb{R}^{1 \times n}$, $k_0(\mathcal{X}_n, \boldsymbol{x}') = (k_0(\boldsymbol{x}', \boldsymbol{x}_i), ..., k_0(\boldsymbol{x}', \boldsymbol{x}_n))^T \in \mathbb{R}^n$, and $\Sigma^n$ is an $n$-dimensional diagonal matrix with noise $\tau^2(\boldsymbol{x}_i)$ being its diagonal elements. Given the posterior GP, the predictive distribution of $y$ can be derived:

$$y(\boldsymbol{x}) \mid \mathcal{X}_n, F_n, \boldsymbol{x} \sim \mathcal{N}(\mu_n(\boldsymbol{x}), \sigma_n^2(\boldsymbol{x})), \tag{7}$$

with $\sigma_n^2(\boldsymbol{x}) = k_n(\boldsymbol{x}, \boldsymbol{x})$.

The noise variance function $\tau$ is unknown but it can be approximated based on observations. Numerous GP variants model input-dependent (heteroscedastic) noise; as noise modeling is not our focus, we do not elaborate further.

### 2.2.2 ACQUISITION FUNCTION

Acquisition functions guide the selection of new sample points. In this section, we briefly introduce two widely used acquisition functions: EI and KG. The EI criterion selects the next point by maximizing the expected improvement over the current best observed value. Let $y^* = \min_{\boldsymbol{x}_i \in \mathcal{X}_n} y(\boldsymbol{x}_i)$ denote the minimum objective function of all samples. The EI at point $\boldsymbol{x}$ is defined as:

$$\text{EI}(\boldsymbol{x}) = \mathbb{E}[\max(y^* - y(\boldsymbol{x}), 0)], \tag{8}$$

where the expectation is taken over the GP posterior distribution at $\boldsymbol{x}$. In noisy settings, $y^*$ is not directly observable. A common practical proxy is the incumbent based on the GP posterior mean,

$$\mu_n^* = \min_{\boldsymbol{x}_i \in \mathcal{X}_n} \mu_n(\boldsymbol{x}_i).$$

Alternatively, some studies treat $y^*$ as a random variable induced by the GP posterior over $y(\cdot)$ and approximate its distribution via Monte Carlo sampling.

The KG acquisition function selects the next sample point by maximizing the expected reduction in the posterior minimum after evaluating the sample. Before sampling, the posterior minimum over the search space is

$$\mu_n^+ = \min_{\boldsymbol{x} \in \mathcal{X}} \mu_n(\boldsymbol{x}).$$

Suppose a noisy evaluation is conducted at a candidate point $\boldsymbol{x}$, yielding an observation $f \sim \mathcal{N}(\mu_n(\boldsymbol{x}), \sigma_n^2(\boldsymbol{x}) + \tau^2(\boldsymbol{x}))$. The GP posterior is then updated to incorporate $(\boldsymbol{x}, f)$, resulting in a new posterior mean function $\mu_{n+1}$, and the posterior minimum after this update is denoted by:

$$\mu_{n+1}^+ = \min_{\boldsymbol{x} \in \mathcal{X}} \mu_{n+1}(\boldsymbol{x}).$$

$\mu_{n+1}^+$ is a random variable conditional on the observation $(\boldsymbol{x}, f)$, and the conventional KG function is defined as the expected reduction in the posterior minimum:

$$\mathbb{E}_f \left[ \mu_n^+ - \mu_{n+1}^+ \mid \boldsymbol{x}_{n+1} = \boldsymbol{x}, f \right]. \tag{9}$$

However, given $\boldsymbol{x}$, calculating the expectation can be computationally intractable because for different realizations of $f$, the minimal of $\mu_{n+1}$ should be calculated over the continuous search space $\mathcal{X}$. One widely used KG function restrict the minimization to the set of sampled points. In the case, the posterior minimum is defined as:

$$\mu_n^* = \min_{\boldsymbol{x}' \in \mathcal{X}_n} \mu_n(\boldsymbol{x}'),$$

and

$$\mu_{n+1}^* = \min_{\boldsymbol{x}' \in \mathcal{X}_n \cup \{\boldsymbol{x}\}} \mu_{n+1}(\boldsymbol{x}').$$

Consequently, the KG is defined as:

$$\text{KG}(\boldsymbol{x}) = \mathbb{E}_f \left[ \mu_n^* - \mu_{n+1}^* \mid \boldsymbol{x}_{n+1} = \boldsymbol{x}, f \right]. \tag{10}$$

BO algorithms iteratively build GP surrogate models and optimize acquisition functions to select new samples. The process continues until the evaluation budget is exhausted. At termination, let $\mathcal{X}_N$ denote the set of sampled points and $\mu_N$ the posterior mean function. The algorithm then recommends the solution

$$\hat{\boldsymbol{x}} = \arg \min_{\boldsymbol{x} \in \mathcal{X}_N} \mu_N(\boldsymbol{x}), \tag{11}$$

that is, the sampled point with the lowest posterior mean.

## 3 AN IDENTIFICATION-AWARE SAMPLING STRATEGY

### 3.1 AN EXAMPLE OF THE IDENTIFICATION ISSUE

The *identification issue* refers to the discrepancy between the truly best solution among the sampled points and the solution that the algorithm identifies as best. We first provide an example of the identification issue.

Suppose that we are to minimize the expected value of the following one-dimensional noisy function over the domain $[-\pi, \pi]$

$$f(x) = \sin(x) + \epsilon(x), \quad \epsilon(x) \sim \mathcal{N}(0, x^2) \tag{12}$$

The BO algorithm is repeated 10 times. Figure 1(a) shows its performance: the blue line is the true best objective among sampled points, and the red line is the value of the solution that would be recommended at each iteration (the sample with the lowest posterior mean). Although the algorithm frequently samples promising candidates, it seldom identifies them as the final recommendation. This gap undermines the practical value of BO, since in real applications only the algorithm's returned solution matters.

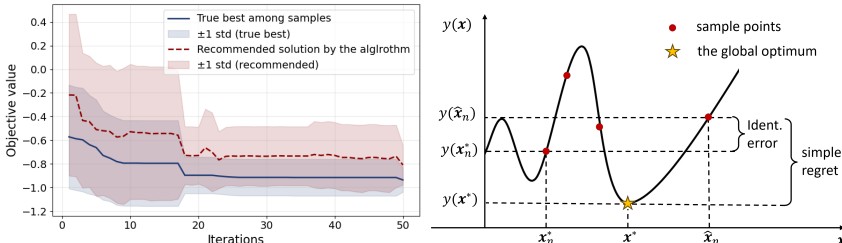

Figure 1: Illustration of the identification issue; (a) Average performance of BO for minimizing a noisy function. The blue curve shows the true best objective among sampled points, while the red curve shows the objective of the algorithm's recommended solution at each iteration.

### 3.2 DEFINITION OF THE IDENTIFICATION ERROR

We first provide the definition of the identification error. Let $\mathcal{X}_n = \{\boldsymbol{x}_1, \boldsymbol{x}_2, \ldots, \boldsymbol{x}_n\} \subset \mathcal{X}$ be the set of $n$ points evaluated by the algorithm. The best solution among all evaluated points is given by

$$\boldsymbol{x}_n^* = \arg\min_{\boldsymbol{x} \in \mathcal{X}_n} y(\boldsymbol{x}), \tag{13}$$

with the corresponding optimal function value

$$y(\boldsymbol{x}_n^*) = \min_{\boldsymbol{x} \in \mathcal{X}_n} y(\boldsymbol{x}). \tag{14}$$

Since $y(\boldsymbol{x})$ is not directly observable, the algorithm typically selects a final recommendation based on the posterior GP mean:

$$\hat{\boldsymbol{x}}_n = \arg\min_{\boldsymbol{x} \in \mathcal{X}_n} \mu_n(\boldsymbol{x}), \tag{15}$$

where $\mu_n(\boldsymbol{x})$ is the posterior mean of GP. The corresponding true performance of the recommended point is $y(\hat{\boldsymbol{x}}_n)$.

We define the identification error as

$$R_n := y(\hat{\boldsymbol{x}}_n) - y(\boldsymbol{x}_n^*). \tag{16}$$

The quantity $R_n$ measures the optimality gap between the best solution contained in the sample set and the one selected by the algorithm based on posterior belief. It is strictly non-negative and equals zero if and only if the algorithm correctly identifies the best solution from the sample set. Notably, this notion differs from the simple regret, which measures the gap between $y(\hat{\boldsymbol{x}}_n)$ and the global optimum $\min_{\boldsymbol{x} \in \mathcal{X}} y(\boldsymbol{x})$ over the entire domain $\mathcal{X}$. Figure 1(b) illustrates the concepts of identification error and simple regret. The black curve denotes the underlying objective function, and the red dots indicate sampled points. The global optimum, the best sampled solution, and the algorithm's recommended solution are highlighted in the figure.

### 3.3 IDENTIFICATION-AWARE ACQUISITION: ONE–STEP BAYES RISK MINIMIZATION

Given the definition of the identification error, this section explores how to control the identification error by generating sample decisions. Suppose that the next sample point is $\boldsymbol{x}_{n+1}$ and the observation on the sample $f_{n+1}$, the identification error is therefore

$$R_{n+1} = y(\hat{\boldsymbol{x}}_{n+1}) - y(\boldsymbol{x}_{n+1}^*) \tag{17}$$

where $\hat{\boldsymbol{x}}_{n+1} = \arg\min_{\boldsymbol{x} \in \mathcal{X}_n \cup \{\boldsymbol{x}_{n+1}\}} \mu_{n+1}(\boldsymbol{x})$, $\boldsymbol{x}_{n+1}^* = \arg\min_{\boldsymbol{x} \in \mathcal{X}_n \cup \{\boldsymbol{x}_{n+1}\}} y(\boldsymbol{x})$. Since $\mu_{n+1}$ is conditional on $f_{n+1}$, $R_{n+1}$ is a random variable. Our key idea is to design a myopic sampling strategy that minimizes the expected $R_{n+1}$ in a one-step lookahead.

We define $m_n := \min_{\boldsymbol{x} \in \mathcal{X}_n} y(\boldsymbol{x})$ and $m_{n+1} := \min_{\boldsymbol{x} \in \mathcal{X}_n \cup \{\boldsymbol{x}_{n+1}\}} y(\boldsymbol{x})$. This leads to the following one–step Bayes risk

$$\mathcal{L}_{n+1}(\boldsymbol{x}_{n+1}) := \mathbb{E}[y(\hat{\boldsymbol{x}}_{n+1}) - m_{n+1} \mid \mathcal{D}_n, \boldsymbol{x}_{n+1}, f_{n+1}], \tag{18}$$

where $\mathcal{D}_n := \{\mathcal{X}_n, F_n\}$ denotes all observations available at step $n$. Since $m_{n+1} = \min\{m_n, y(\boldsymbol{x})\}$, we can decompose Eq. (18) as

$$\begin{aligned}
\mathcal{L}_{n+1}(\boldsymbol{x}_{n+1}) &= \mathbb{E}[y(\hat{\boldsymbol{x}}_{n+1}) - \mathbb{I}\{m_n > y(\boldsymbol{x}_{n+1})\}(y(\boldsymbol{x}_{n+1}) - m_n) - m_n \mid \mathcal{D}_n, \boldsymbol{x}_{n+1}, f_{n+1}] \\
&= \mathbb{E}[y(\hat{\boldsymbol{x}}_{n+1}) \mid \mathcal{D}_n, \boldsymbol{x}_{n+1}, f_{n+1}] + \mathbb{E}[(m_n - y(\boldsymbol{x}_{n+1}))_+ \mid \mathcal{D}_n, \boldsymbol{x}_{n+1}, f_{n+1}] - m_n,
\end{aligned} \tag{19}$$

where $(\cdot)_+ = \max(\cdot, 0)$. The second term is precisely the *expected improvement* as defined in Eq. (8)

$$\mathrm{EI}(\boldsymbol{x}_{n+1}) = \mathbb{E}[(m_n - y(\boldsymbol{x}_{n+1}))_+ \mid \mathcal{D}_n, \boldsymbol{x}_{n+1}, f_{n+1}].$$

The first term in Eq. (19) is the expected objective value of the recommended solution after observing $(\boldsymbol{x}_{n+1}, f_{n+1})$. Under the decision rule $\hat{\boldsymbol{x}}_{n+1} = \arg\min_{\boldsymbol{x} \in \mathcal{X}_n \cup \{\boldsymbol{x}_{n+1}\}} \mu_{n+1}(\boldsymbol{x})$, the first term in Eq. (19) can be approximated by

$$\begin{aligned}
\mathbb{E}[y(\hat{\boldsymbol{x}}_{n+1}) \mid \mathcal{D}_n, \boldsymbol{x}_{n+1}, f_{n+1}] &= \mathbb{E}_{f_{n+1}}\Big[\min_{\boldsymbol{x}' \in \mathcal{X} \cup \{\boldsymbol{x}_{n+1}\}} \mu_{n+1}(\boldsymbol{x}') \mid \mathcal{D}_n, \boldsymbol{x}_{n+1}, f_{n+1}\Big] \\
&= \mathbb{E}_{f_{n+1}}\Big[\mu_{n+1}^* - \mu_n^* + \mu_n^* \mid \mathcal{D}_n, \boldsymbol{x}_{n+1}, f_{n+1}\Big] \\
&= -\mathrm{KG}(\boldsymbol{x}_{n+1}) + \mu_n^*
\end{aligned} \tag{20}$$

where $\mu_n^* = \min_{\boldsymbol{x} \in \mathcal{X}_n} \mu_n(\boldsymbol{x})$ is a constant and $\mu_n^* = \min_{\boldsymbol{x} \in \mathcal{X}_n \cup \{\boldsymbol{x}_{n+1}\}} \mu_{n+1}(\boldsymbol{x})$

Since $m_n$ does not depend on $\boldsymbol{x}_{n+1}$, the one–step Bayes risk can be represented by

$$\mathcal{L}_{n+1}(\boldsymbol{x}_{n+1}) = m_n + \mathrm{EI}(\boldsymbol{x}_{n+1}) + \left[\mu_n^* - \mathrm{KG}(\boldsymbol{x}_{n+1})\right].$$

Since both $m_n$ and $\mu_n^*$ are independent of $\boldsymbol{x}_{n+1}$, minimizing the Bayes risk is equivalent to minimize:

$$\mathrm{EI}(\boldsymbol{x}_{n+1}) - \mathrm{KG}(\boldsymbol{x}_{n+1}) \tag{21}$$

This leads to a rather striking observation: if one seeks to myopically minimize the identification error, the required acquisition rule reduces to the simple difference between two widely used criteria, KG and EI. This rule is easy to implement, benefits from existing theoretical results, and has a clear interpretation—KG improves the recommended solution, EI improves the best sampled solution, and their difference captures the reduction in identification error. Taking their difference captures the intuition that identification error decreases when the recommended solution improves substantially while the best sampled solution shows little or no improvement.

However, minimizing identification error alone is insufficient, as it does not guarantee improvement of the best solution. Hence, the sampling strategy should balance two goals: (i) improving the best sampled solution (EI) and (ii) reducing identification error (EI–KG). Combining these with appropriate weights leads to our new acquisition function:

$$\begin{aligned}
\boldsymbol{x}_{n+1} &= \arg\max_{\boldsymbol{x} \in \mathcal{X}} \quad \alpha_n(\mathrm{KG}(\boldsymbol{x}) - \mathrm{EI}(\boldsymbol{x})) + \mathrm{EI}(\boldsymbol{x}) \\
&= \arg\max_{\boldsymbol{x} \in \mathcal{X}} \quad \alpha_n \mathrm{KG}(\boldsymbol{x}) + (1 - \alpha_n)\mathrm{EI}(\boldsymbol{x}).
\end{aligned} \tag{22}$$

where $\alpha_n > 0$ denotes the weight of identification error in making sampling decisions. As more observations accumulate, $\alpha_n$ is gradually increased, shifting the focus towards reducing the identification error.

**Remark 1:** A natural design of $\alpha_n$ is to prioritize exploration of promising solutions in the early stage ($\alpha = 0$, corresponding to EI) and gradually shift toward emphasizing identification in the later stage ($\alpha = \infty$, corresponding to KG–EI). To capture this transition, we model the evolution of $\alpha_n$ using an exponential function,

$$\alpha_n = \beta \exp(\lambda n) - \beta, \tag{23}$$

where $\beta > 0$ and $\lambda > 0$ control the growth rate and scale.

# 4 CONVERGENCE ANALYSIS

We analyze the convergence of the proposed acquisition function when it is incorporated into the BO framework. The GP surrogate provides posterior mean $\mu_{n-1}(\boldsymbol{x})$ and variance $\sigma_{n-1}^2(\boldsymbol{x})$ after $n-1$ observations.

**Confidence band.** Assume that the objective function $y$ lies in the reproducing kernel Hilbert space (RKHS) of the kernel $k_0$ with bounded norm. Following Chowdhury & Gopalan (2017); Makarova et al. (2021), for any $\delta \in (0, 1)$ there exists a sequence $\{\beta_n\}$ such that, with probability at least $1-\delta$,

$$\big| y(\boldsymbol{x}) - \mu_{n-1}(\boldsymbol{x}) \big| \leq \beta_n \, \sigma_{n-1}(\boldsymbol{x}), \qquad \forall \boldsymbol{x} \in \mathcal{X}, \ \forall n. \tag{24}$$

Here $\beta_n$ depends on the RKHS norm bound, the confidence level $\delta$, and the maximal information gain $\gamma_{n-1}$, and serves as the confidence width. Given the confidence band, we can derive the following upper bound

**Lemma 1.** *Let $\boldsymbol{x}^\star = \arg\min_{\boldsymbol{x}\in\mathcal{X}} y(\boldsymbol{x})$ denote the global optimum, and define the instantaneous regret as*

$$r_n := y(\boldsymbol{x}^\star) - y(\boldsymbol{x}_n).$$

*On the confidence event (24), we have*

$$r_n \leq I_n(\boldsymbol{x}^\star) + 2\beta_n \, \sigma_{n-1}(\boldsymbol{x}_n) + 2\beta_n \, \sigma_{n-1}(\boldsymbol{x}^\star), \tag{25}$$

*where*

$$I_n(\boldsymbol{x}) := \Big[ \min_{\boldsymbol{x}'\in\mathcal{X}_{n-1}} \mu_{n-1}(\boldsymbol{x}') - y(\boldsymbol{x}) \Big]_+$$

*denotes the realised improvement of $\boldsymbol{x}$ relative to the current posterior best.*

**Lemma 2.** *For any $\boldsymbol{x}$, the following inequalities hold:*

$$I_n(\boldsymbol{x}) - (\beta_n + c_0) \, \sigma_{n-1}(\boldsymbol{x}) \leq \mathrm{EI}_n(\boldsymbol{x}) \leq I_n(\boldsymbol{x}) + (\beta_n + c_0) \, \sigma_{n-1}(\boldsymbol{x}), \tag{26}$$

*where $c_0 = \sqrt{2/\pi}$ and $\mathrm{EI}_n$ denotes the EI function established based on n-1 evaluations.*

Given the results of **Lemma 2**, we can further derive the upper bound of the instantaneous regret,

**Proposition 1.** *Let $A_n(\boldsymbol{x}) = \alpha_n \mathrm{KG}_n(\boldsymbol{x}) + (1 - \alpha_n)\mathrm{EI}_n(\boldsymbol{x})$ be the blended acquisition. On the confidence event (24), the instantaneous regret satisfies*

$$r_n \leq (C_{\mathrm{EI}} + C_{\mathrm{KG}} + 2\beta_n) \, \sigma_{n-1}(\boldsymbol{x}_n) + (3\beta_n + c_0) \, \sigma_{n-1}(\boldsymbol{x}^\star), \tag{27}$$

*where $C_{\mathrm{EI}}, C_{\mathrm{KG}} > 0$ are absolute constants such that*

$$\mathrm{EI}_n(\boldsymbol{x}) \leq C_{\mathrm{EI}} \, \sigma_{n-1}(\boldsymbol{x}), \quad \mathrm{KG}_n(\boldsymbol{x}) \leq C_{\mathrm{KG}} \, \sigma_{n-1}(\boldsymbol{x}) \qquad \forall \boldsymbol{x} \in \mathcal{X}, \ \forall n.$$

**Proposition 1** follows standard results in BO analysis, where both EI and KG can be bounded by posterior standard deviation up to universal constants (see Bull (2011); Frazier et al. (2009)).

**Theorem 1.** *Let $R_N = \sum_{n=1}^N r_n$ denote the cumulative regret after $N$ rounds. On the confidence event (24), with probability at least $1 - \delta$ we have*

$$R_N \leq (C_{\mathrm{EI}} + C_{\mathrm{KG}} + 2\beta_N) \sqrt{\frac{2N\gamma_N}{\log(1+\tau_{\max}^{-2})}} + (3\beta_N + c_0) \, \sigma_0(\boldsymbol{x}^\star)\sqrt{N},$$

*where $\gamma_N$ is the maximum information gain that depends on the kernel function and $\tau_{\max}$ is an upper bound on the noise variance. The constants $C_{\mathrm{EI}}$ and $C_{\mathrm{KG}}$ arise from linear variance bounds for the EI and KG, respectively, while $c_0 = \sqrt{2/\pi}$ is a universal Lipschitz constant controlling the deviation between realised and expected improvement.*

## 5 EXPERIMENTS AND RESULTS

### 5.1 EXPERIMENTAL SETUP

Following Jalali et al. (2017), we evaluate two 2-D benchmarks: the six-hump Camelback and the rescaled Branin, each perturbed with heteroscedastic Gaussian noise $\tau(\boldsymbol{x}) = a(y(\boldsymbol{x}) + b)$. We consider two noise structures—Best (minimum variance at the global optimum) and Worst (maximum variance at the global optimum)—and two magnitudes, Light and Heavy, with parameters $(a, b)$ specified in the appendix. To avoid bias in acquisition optimization, each domain is discretized into 100 candidate points using Latin hypercube sampling. We compare BO with knowledge gradient (KG) (Scott et al., 2011), BO with noisy expected improvement (NEI) (Letham et al., 2019), and BO with our proposed acquisition in two variants: (i) an identification-only rule, and (ii) the full Identification-Error Aware Acquisition (IDEA).

Each problem is initialized with 20 space-filling points ($10d$ rule (Jones et al., 1998)), each evaluated 200 times to estimate local noise. The sample mean serves as the observation, and the variance divided by 200 as the noise estimate. Two GPs are fitted: one on the means (objective surrogate) and one on the log noise variance (noise surrogate). BO then runs 100 iterations on a 100-point grid, with each algorithm repeated 10 times under different seeds. Experiments were conducted on a PC with an Intel® Core™ Ultra 9 185H CPU.

Recall that at iteration $n$, we denote by $\boldsymbol{x}_n^* = \arg\min_{\boldsymbol{x} \in \mathcal{X}_n} y(\boldsymbol{x})$ the best solution among the sampled points, and by $\hat{\boldsymbol{x}}_n = \arg\min_{\boldsymbol{x} \in \mathcal{X}_n} \mu_n(\boldsymbol{x})$ the solution that would be returned if the algorithm were terminated. Let $\boldsymbol{x}^* = \arg\min_{\boldsymbol{x} \in \mathcal{X}} y(\boldsymbol{x})$ denote the global optimum. We evaluate each algorithm using three performance metrics: **Simple regret:** $y(\hat{\boldsymbol{x}}_n) - y(\boldsymbol{x}^*)$ that measures the performance of the returned solution relative to the global optimum; **Identification error:** $y(\hat{\boldsymbol{x}}_n) - y(\boldsymbol{x}_n^*)$ that measures the discrepancy between the returned solution and the best sampled solution; **Best-observed regret:** $y(\boldsymbol{x}_n^*) - y(\boldsymbol{x}^*)$ that measures the algorithm's ability to discover promising solutions.

### 5.2 RESULTS AND DISCUSSIONS

We first evaluate the identification error of each method. Figure 2 reports the identification error across iterations for each test function and scenario. As expected, using KG–EI as the acquisition function substantially reduces identification error. Moreover, the proposed IDEA acquisition function further improves performance, showing the strongest ability to eliminate identification error.

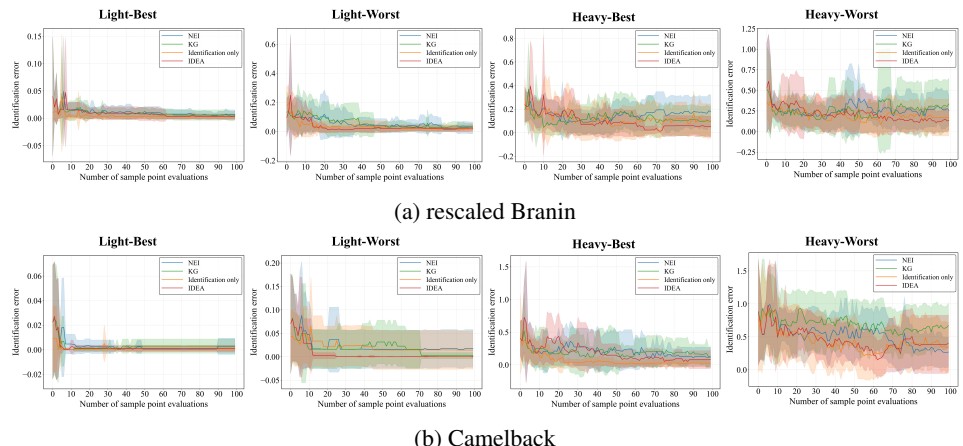

Figure 2: Identification errors of different methods across test scenarios

Figure 3 presents the simple regret of each algorithm. This metric is particularly important because the solution returned by an algorithm is what will ultimately be implemented in practice; hence, simple regret reflects practical performance. The results show that IDEA consistently outperforms the benchmark methods. Notably, under heavy noise, simple regret nearly coincides with identifi-

cation error, indicating that identification error is the primary factor limiting the performance of BO algorithms in noisy settings.

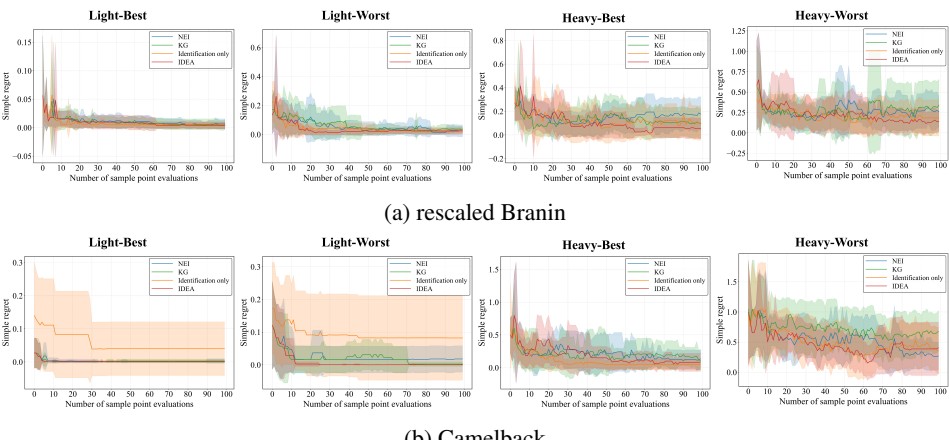

(a) rescaled Branin

(b) Camelback

Figure 3: Simple regrets of different methods across test scenarios

Finally, we examine the best-observed regret, which measures the ability of algorithms to discover promising solutions during the search process. The NEI criterion achieves the best performance in this regard. In contrast, when KG–EI is applied, the algorithm tends not to identify better solutions. This is because focusing solely on minimizing identification error encourages repeated evaluations of existing candidates or the selection of inferior points, rather than exploration of potentially superior ones.

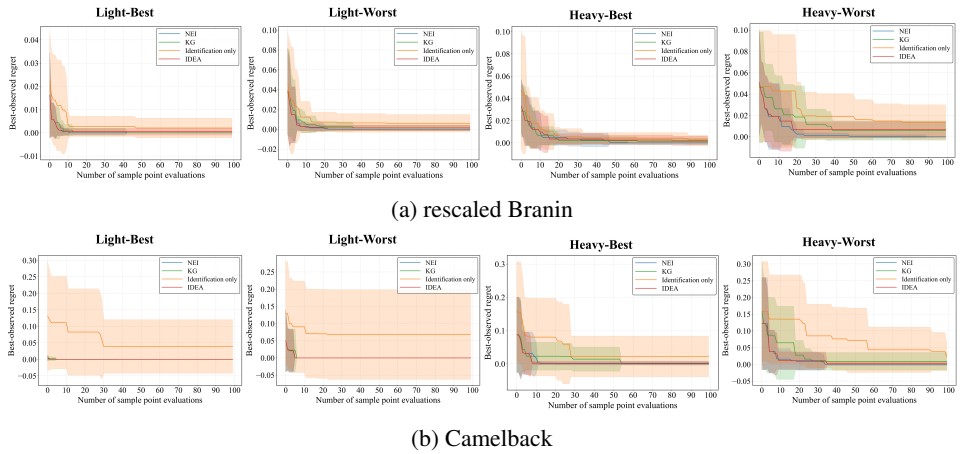

(a) rescaled Branin

(b) Camelback

Figure 4: Best-observed regrets of different methods across test scenarios

## 6 CONCLUSION

This paper investigates the identification problem in BO and provides the first principled treatment within a standard BO framework. We formally defined identification error and derived a myopic acquisition rule that directly minimizes it, revealing a surprising equivalence to the difference between KG and EI. Building on this insight, we proposed the IDEA function, which balances exploration, exploitation, and identification. We established its asymptotic no-regret guarantee and demonstrated its effectiveness on benchmark functions with heteroscedastic noise.

Our results highlight that reliable final solution identification is as critical as exploration–exploitation trade-offs in practical BO applications. We believe this work opens up several promising directions, including extending IDEA to high-dimensional settings, integrating it with scalable surrogate models, and applying it to real-world problems such as AutoML and resource allocation.

ETHICS STATEMENT

This work raises no specific ethical concerns. All experiments were conducted on publicly available benchmark problems or synthetic data, without the use of personally identifiable information, sensitive data, or human subjects. We have carefully considered potential societal impacts and found no direct risks beyond the intended scope of optimization and machine learning research.

REPRODUCIBILITY STATEMENT

To ensure reproducibility, we provide complete details of the experimental settings, including problem definitions, initialization rules, and hyperparameter configurations. All algorithms were implemented with publicly available libraries. Upon publication, we will release our code, random seeds, and processed datasets to facilitate independent verification and reuse.

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

# A APPENDIX

## A.1 PROOF

### A.1.1 PROOF OF LEMMA 1

By definition of the instantaneous regret,

$$r_n = y(\boldsymbol{x}^\star) - y(\boldsymbol{x}_n).$$

We first insert the posterior mean terms in order to exploit the confidence bound (24):

$$r_n = \big(y(\boldsymbol{x}^\star) - \mu_{n-1}(\boldsymbol{x}^\star)\big) + \big(\mu_{n-1}(\boldsymbol{x}^\star) - \mu_{n-1}(\boldsymbol{x}_n)\big) + \big(\mu_{n-1}(\boldsymbol{x}_n) - y(\boldsymbol{x}_n)\big).$$

On the confidence event, the first and third brackets are bounded by

$$|y(\boldsymbol{x}^\star) - \mu_{n-1}(\boldsymbol{x}^\star)| \le \beta_n \sigma_{n-1}(\boldsymbol{x}^\star), \qquad |\mu_{n-1}(\boldsymbol{x}_n) - y(\boldsymbol{x}_n)| \le \beta_n \sigma_{n-1}(\boldsymbol{x}_n).$$

Therefore,

$$r_n \le \mu_{n-1}(\boldsymbol{x}^\star) - \mu_{n-1}(\boldsymbol{x}_n) + \beta_n \sigma_{n-1}(\boldsymbol{x}^\star) + \beta_n \sigma_{n-1}(\boldsymbol{x}_n).$$

Next, we decompose the mean difference via the current posterior minimum $\mu_{n-1}^* = \min_{\boldsymbol{x}' \in \mathcal{X}_{n-1}} \mu_{n-1}(\boldsymbol{x}')$:

$$\mu_{n-1}(\boldsymbol{x}^\star) - \mu_{n-1}(\boldsymbol{x}_n) = \big(\mu_{n-1}(\boldsymbol{x}^\star) - \mu_{n-1}^*\big) + \big(\mu_{n-1}^* - \mu_{n-1}(\boldsymbol{x}_n)\big).$$

For the second bracket, apply again the confidence bound at $\boldsymbol{x}_n$:

$$\mu_{n-1}^* - \mu_{n-1}(\boldsymbol{x}_n) \le \mu_{n-1}^* - y(\boldsymbol{x}_n) + \beta_n \sigma_{n-1}(\boldsymbol{x}_n).$$

For the first bracket, compare with the true value at $\boldsymbol{x}^\star$:

$$\mu_{n-1}(\boldsymbol{x}^\star) - \mu_{n-1}^* \leq y(\boldsymbol{x}^\star) - \mu_{n-1}^* + \beta_n \sigma_{n-1}(\boldsymbol{x}^\star).$$

Now recall the definition

$$I_n(\boldsymbol{x}^\star) := [\,\mu_{n-1}^* - y(\boldsymbol{x}^\star)\,]_+.$$

Using the identity $a = [a]_+ - [-a]_+$ with $a = \mu_{n-1}^* - y(\boldsymbol{x}^\star)$, we can rewrite

$$y(\boldsymbol{x}^\star) - \mu_{n-1}^* = -I_n(\boldsymbol{x}^\star) + [\,y(\boldsymbol{x}^\star) - \mu_{n-1}^*\,]_+.$$

Hence the upper bound above produces both $I_n(\boldsymbol{x}^\star)$ and an additional $\beta_n \sigma_{n-1}(\boldsymbol{x}^\star)$. Collecting all terms, we conclude that

$$r_n \;\leq\; I_n(\boldsymbol{x}^\star) + 2\beta_n \sigma_{n-1}(\boldsymbol{x}_n) + 2\beta_n \sigma_{n-1}(\boldsymbol{x}^\star),$$

which is the desired result.

### A.1.2    PROOF OF LEMMA 2

Let $a = \mu_{n-1}^* = \min_{\boldsymbol{x}' \in \mathcal{X}_{n-1}} \mu_{n-1}(\boldsymbol{x}')$ and denote by $Y \sim \mathcal{N}(\mu_{n-1}(\boldsymbol{x}), \sigma_{n-1}^2(\boldsymbol{x}))$ the GP predictive distribution. Define $\phi(u) = (a - u)_+$. Then by definition

$$\mathrm{EI}_n(\boldsymbol{x}) = \mathbb{E}[\phi(Y)], \qquad I_n(\boldsymbol{x}) = \phi(y(\boldsymbol{x})).$$

We wish to control the deviation $|\mathrm{EI}_n(\boldsymbol{x}) - I_n(\boldsymbol{x})|$. Introduce $\phi(\mu_{n-1}(\boldsymbol{x}))$ as an intermediate anchor:

$$|\mathrm{EI}_n(\boldsymbol{x}) - I_n(\boldsymbol{x})| \leq |\mathrm{EI}_n(\boldsymbol{x}) - \phi(\mu_{n-1}(\boldsymbol{x}))| + |\phi(\mu_{n-1}(\boldsymbol{x})) - \phi(y(\boldsymbol{x}))|.$$

For the first term, since $\phi$ is 1-Lipschitz,

$$|\mathrm{EI}_n(\boldsymbol{x}) - \phi(\mu_{n-1}(\boldsymbol{x}))| \leq \mathbb{E}|\phi(Y) - \phi(\mu_{n-1}(\boldsymbol{x}))| \leq \mathbb{E}|Y - \mu_{n-1}(\boldsymbol{x})|.$$

Because $Y$ is Gaussian, this expectation is $c_0 \sigma_{n-1}(\boldsymbol{x})$ with $c_0 = \sqrt{2/\pi}$.

For the second term, by the Lipschitz property of $\phi$,

$$|\phi(\mu_{n-1}(\boldsymbol{x})) - \phi(y(\boldsymbol{x}))| \leq |\mu_{n-1}(\boldsymbol{x}) - y(\boldsymbol{x})| \leq \beta_n \sigma_{n-1}(\boldsymbol{x})$$

on the confidence event (24).

Summing the two bounds gives

$$|\mathrm{EI}_n(\boldsymbol{x}) - I_n(\boldsymbol{x})| \leq (\beta_n + c_0)\,\sigma_{n-1}(\boldsymbol{x}),$$

which yields the stated two-sided inequality.

### A.1.3    PROOF OF PROPOSITION 1

From **Lemma 1** we have

$$r_n \;\leq\; I_n(\boldsymbol{x}^\star) \;+\; 2\beta_n \sigma_{n-1}(\boldsymbol{x}_n) \;+\; 2\beta_n \sigma_{n-1}(\boldsymbol{x}^\star).$$

It remains to upper bound $I_n(\boldsymbol{x}^\star)$. By **Lemma 2**,

$$I_n(\boldsymbol{x}^\star) \;\leq\; \mathrm{EI}_n(\boldsymbol{x}^\star) + (\beta_n + c_0)\sigma_{n-1}(\boldsymbol{x}^\star).$$

Since $\boldsymbol{x}_n$ maximises $A_n$, we have

$$A_n(\boldsymbol{x}_n) \;\geq\; A_n(\boldsymbol{x}^\star) \;\geq\; (1 - \alpha_n)\mathrm{EI}_n(\boldsymbol{x}^\star),$$

which implies

$$\mathrm{EI}_n(\boldsymbol{x}^\star) \;\leq\; \tfrac{1}{1-\alpha_n} A_n(\boldsymbol{x}_n).$$

Furthermore, the acquisition value at $\boldsymbol{x}_n$ can be bounded by

$$A_n(\boldsymbol{x}_n) \;\leq\; \alpha_n C_{\mathrm{KG}}\sigma_{n-1}(\boldsymbol{x}_n) + (1 - \alpha_n)C_{\mathrm{EI}}\sigma_{n-1}(\boldsymbol{x}_n).$$

Combining these inequalities gives

$$I_n(\boldsymbol{x}^\star) \;\leq\; (C_{\mathrm{EI}} + C_{\mathrm{KG}})\sigma_{n-1}(\boldsymbol{x}_n) + (\beta_n + c_0)\sigma_{n-1}(\boldsymbol{x}^\star).$$

Substituting into the regret decomposition completes the proof of **Proposition 1**.    $\square$

### A.1.4 PROOF OF THEOREM 1

Summing the inequality of **Proposition 1** over $n = 1, \ldots, N$,

$$R_N = \sum_{n=1}^{N} r_n \leq (2\beta_N + C_{EI} + C_{KG}) \sum_{n=1}^{N} \sigma_{n-1}(\boldsymbol{x}_n) + (3\beta_N + c_0) \sum_{n=1}^{N} \sigma_{n-1}(\boldsymbol{x}^\star).$$

For the first term, apply Cauchy–Schwarz and the information gain bound (Chowdhury & Gopalan, 2017; Makarova et al., 2021):

$$\sum_{n=1}^{N} \sigma_{n-1}(\boldsymbol{x}_n) \leq \sqrt{N \sum_{n=1}^{N} \sigma_{n-1}^2(\boldsymbol{x}_n)} \leq \sqrt{\frac{2N\gamma_N}{\log(1+\tau_{\max}^{-2})}}.$$

For the second term, note $\sigma_{n-1}(\boldsymbol{x}^\star) \leq \sigma_0(\boldsymbol{x}^\star)$ for all $n$, hence

$$\sum_{n=1}^{N} \sigma_{n-1}(\boldsymbol{x}^\star) \leq N^{1/2} \sigma_0(\boldsymbol{x}^\star) \sqrt{N}.$$

Combining the two bounds yields

$$R_N \leq (2\beta_N + C_{EI} + C_{KG}) \sqrt{\frac{2N\gamma_N}{\log(1+\tau_{\max}^{-2})}} + (3\beta_N + c_0) \sigma_0(\boldsymbol{x}^\star) \sqrt{N}.$$

Finally, applying standard bounds for the information gain $\gamma_N$ gives the stated rates for squared exponential and Matérn kernels.

### A.2 BENCHMARK FUNCTIONS AND NOISE DESIGN

### A.2.1 ANALYTICAL TEST FUNCTIONS

We consider two standard 2-D benchmarks.

**Six-hump Camelback.**

$$f(x_1, x_2) = 4x_1^2 - 2.1x_1^4 + \tfrac{1}{3}x_1^6 + x_1 x_2 - 4x_2^2 + 4x_2^4, \qquad -2 \leq x_1 \leq 2, \ -1 \leq x_2 \leq 1.$$

The global minimizer and value are approximately $x^\star = (0.0977, -0.6973)$, $f(x^\star) = -1.0294$. The response range on the domain is $R_f \approx 7.3$.

**Rescaled Branin.** Let $\bar{x}_1 = 15x_1 - 5$ and $\bar{x}_2 = 15x_2$ for $(x_1, x_2) \in [0, 1]^2$. Define

$$f(x_1, x_2) = \frac{1}{51.95}\left[\left(\bar{x}_2 - \frac{5.1}{4\pi^2}\bar{x}_1^2 + \frac{5}{\pi}\bar{x}_1 - 6\right)^2 + \left(10 - \frac{10}{8\pi}\right)\cos(\bar{x}_1) - 44.81\right].$$

The global minimizer and value are approximately $x^\star = (0.541, 0.1348)$, $f(x^\star) = -1.0459$. The response range on the domain is $R_f \approx 6$.

### A.2.2 HETEROSCEDASTIC NOISE MODEL

We perturb each function with input-dependent Gaussian noise. A single noisy observation at $x$ is $\tilde{f}(x) = f(x) + \varepsilon(x)$ with $\varepsilon(x) \sim \mathcal{N}\big(0, \tau^2(x)\big)$. Following the linear-noise scheme, we set

$$\tau(x) = a\big(f(x) + b\big),$$

and calibrate $a, b$ to realize two *noise magnitudes* and two *noise structures*:

- **Magnitude.** *Light*: $\min_x \tau(x) = 0.15 R_f$, $\max_x \tau(x) = 0.6 R_f$. *Heavy*: $\min_x \tau(x) = 1.5 R_f$, $\max_x \tau(x) = 6 R_f$.
- **Structure.** *Best*: noise is smallest at the global minimum and largest at the maximum. *Worst*: noise is largest at the global minimum and smallest at the maximum.

Closed-form calibration yields the $(a, b)$ pairs in Table 1, used in all experiments.

Table 1: Parameters $(a, b)$ in $\tau(x) = a\big(f(x) + b\big)$ by function and scenario.

| Scenario | Six-hump Camelback | Rescaled Branin |
|---|---|---|
| Best–Light | $a = 0.45,\ b = 3.46$ | $a = 0.45,\ b = 3.05$ |
| Best–Heavy | $a = 4.50,\ b = 3.46$ | $a = 4.50,\ b = 3.05$ |
| Worst–Light | $a = -0.45,\ b = -8.704$ | $a = -0.45,\ b = -6.95$ |
| Worst–Heavy | $a = -4.50,\ b = -8.704$ | $a = -4.50,\ b = -6.95$ |

## USE OF LLMS

Large Language Models (LLMs), specifically ChatGPT, were used solely for language polishing and minor stylistic refinement of the manuscript. All technical content, mathematical formulations, algorithm design, and experimental results were developed and validated independently by the authors.

