# OpenReview forum: "Beyond Exploration–Exploitation: An Identification-Aware Bayesian Optimization Method under Noisy Evaluations"
_ICLR.cc/2026/Conference — ICLR 2026 Conference Desk Rejected Submission_

### Official Review · Reviewer_jwEd · 2025-10-21

**Soundness:** 1
**Presentation:** 1
**Contribution:** 2
**Rating:** 2
**Confidence:** 5

**Summary:**

The paper studies a problem of identification-aware Bayesian Optimisation. It is essentially the best-arm identification problem from the multi-armed bandits community, but framed in the context of Bayesian Optimisation (Gaussian Process bandits). That is, they focus on studying the problem of making sure the algorithm at the end of its runtime, is able to identify the point as close as possible to optimal, rather than simply focusing on the best regret. Authors show that minimising the one-step Bayes risk of misidentifying the optimal point is equivalent to minimising the difference between Expected Improvement and Knowledge gradient. Based on this observation, authors propose an algorithm and evaluate it on number of synthetic benchmarks.

**Strengths:**

The result that minimising one-step Bayes risk is equivalent to minimising difference of EI and KG, is interesting and insightful.

**Weaknesses:**

**Significance unclear**

The authors essentially focus on a different problem setting than the one studied in classical BO and even later on page 9 they admit their algorithm underperforms to existing baseline in terms of the standard metric of best regret because, quoting the authors:
> This is because focusing solely on minimizing identification error encourages repeated evaluations
of existing candidates or the selection of inferior points, rather than exploration of potentially superior ones.

So clearly, the objective in the adopted problem setting does not need to coincide with the best regret objective of classical BO. As such, I would like authors to demonstrate on practical problems, why should we care about best point identification rather than best regret. In the applications of BO, I am aware of, best regret is the metric of interest, e.g. in hyperparameter tuning, we simply wish to find a configuration of hyperparameters that performs best, and we do not care if the algorithm queries suboptimal points after finding the optimum, as long as the best solution found has a good objective value.

We could speculate that maybe in some practical problem settings, your objective would could more applicable, but this need to be demonstrated by the authors. As such, for now I am not convinced that the proposed problem setting is relevant.

**Error in convergence proof**

Unfortunately, I believe Theorem 1 is wrong. The error is in transitioning from the penultimate to the last line in the proof in A.1.4. The bound on $\sum_{n=1}^N \sigma_n(x^\star)$ reads $N^{1/2} \sigma_0(x^\star) \sqrt{N}$ — in other words, $N \sigma_0(x^\star)$ (just written in a very confusing way for some reason). But then when combining the bounds in the final line, you lose the $N^{1/2}$ factor, that is the second term should read $(3\beta_N + c_0)\sigma_0(x^\star)N$ (instead of $\sqrt{N}$), **making the bound linear and thus trivial**. An algorithm with linear regret does no better than just a naive strategy of constantly selecting the same point, as such, linear regret means the algorithm **does not converge**. This proof must be corrected, or the claims about convergence must be removed.

**Experiments unconvincing**

It is not clear if the shaded areas are standard errors, standard deviations or CIs based on those. In almost all experiments, the shaded areas overlap meaning that the differences between the algorithms are not statistically significant. The authors initialise the algorithm by sampling 20 points and querying each 200 times to estimate noise, that is a total of 4000 queries to the blackbox, that typically already exceeds the budgets we have in BO, raising concerns as to how realistic this problem setting really is. The plots in the experiments section are too small to be realistically readable.

**Issues with notation**

The notation is completely inconsistent. First authors define the black-box function as $f(x, \xi)$ in Equation (1), then they switch to calling the function value $y(x)$ throughout most of theoretical derivations. Then in equation (12) they defined noisy function as $f(x) = sin(x) + \epsilon(x)$, where $\epsilon(x) \sim \mathcal{N}(0, x^2)$, so revert to calling it $f$, but for some reason omit the explicit dependance on randomness $\xi$, even though function clearly depends on stochastic noise. Authors keep switching between capital $N$ and $n$ when referring to number of samples. Sometimes authors call the kernel function and prior mean as $k$ and $\mu$, and sometimes $k_0$ and $\mu_0$.


**Minor**

Line 294, there are two equations for $\mu_n^\star$, I believe the second one should be $\mu_{n+1}^\star$.

**Questions:**

- What do the shaded areas in the plots represent? Are they standard errors/deviations or CIs based on them or some other quantity?
- The reliance of the proposed algorithm on an additional hyperparameter $\alpha_n$ or $\lambda$ and $\beta$ seems like a major limitation. How were they selected in your experiments?

---

### Official Review · Reviewer_SL9F · 2025-10-26

**Soundness:** 2
**Presentation:** 3
**Contribution:** 1
**Rating:** 2
**Confidence:** 4

**Summary:**

The paper considers the problem of Bayesian Optimization, where the objective is to optimize a black-box function using a fixed budget of queries. Within this framework, the paper focuses on the "identification problem" and aims to solve it by minimizing the identification error. The identification error is defined to be sub-optimality gap between the final output of the algorithm and best point in the query set. The authors show that myopic/one-step minimization of the identification error is equivalent to minimizing a difference between expected improvement (EI) and knowledge gradient (KG), providing a tractable solution to this problem. The authors also propose an algorithm that incorporates "Identification-Error Aware Acquisition" by taking a weighted mean of EI - KG and KG. Theoretical guarantees along with some numerical studies to support the theoretical guarantees are provided by the authors.

**Strengths:**

The paper is clearly written and easy to follow. The problem is clearly stated and the proposed solution is clearly explained.

**Weaknesses:**

A glaring weakness of this paper, in my opinion, is its motivation.

- Why is identification-awareness that important? Why is it important to return the "best" point among the set of queried points?
- I agree with the authors that BO algorithms likely don't return the "best" query point. The authors seem to claim this is some "prevalant weakness" among BO algorithms. I don't see how or why this is a weakness as long the returned point behaves similar enough to the "best" point?
- From a theoretical viewpoint, there are numerous algorithms whose performance in terms of simple regret is well-known. Since simple regret is an upper on identification error, these bounds on existing algorithms can immediately be extended to that on identification error. Based on that, we can easily conclude that the bounds on identification error provided by this algorithm are sub-optimal (since optimal algorithms are known). Moreover, the analysis techniques are well-known in the literature. So, clearly there is no obvious theoretical benefit of this approach over existing ones.
- From a practical point of view, the numerical studies don't suggest any statistically significant advantage of the proposed approach over NEI or KG. So with no theoretical/practical benefit, I am struggling to see the motivation/need for an identification-aware approach.

**Questions:**

In addition to the above questions, I have some additional questions:

- How do arrive at Eqn 20 from Eqn 19? It is stated that there is an approximation. What is the approximation and how reasonable is it?
- In proof of Theorem 1, in how do you arrive at line 667 from from 662-663? Shouldn't the RHS be $N\sigma_0(x^{\star})$ instead of $\sqrt{N} \sigma_0(x^{\star})$? If so, the regret is a constant, unlike what is claimed in the Theorem.
- Can the authors justify why is reasonable to assume that $\tau$ is known or can be estimated? It is required to construct the posterior mean $\mu$.
- Even if $\tau$ is known, the equation in lines 657-658 needs to appropriately modified to account for heteroscadastic noise, especially when $\tau$ is very small. The result as stated only holds for homogeneous noise.
- What is the choice of $\alpha_n$ or equivalently $\beta$, $\lambda$ for the experiments?
- From the description in lines 391-393, it seems the authors have used 4000 queries to estimate the mean and variance and they don't count towards the query budget. Am I understanding this correct? If so, this does not seem fair.

---

### Official Review · Reviewer_MmcK · 2025-10-31

**Soundness:** 1
**Presentation:** 2
**Contribution:** 2
**Rating:** 2
**Confidence:** 3

**Summary:**

The authors claim to contribute:
- A formal definition of identification error in BO
- An acquisition function that balances minimizing identification error with improving the best solution (which includes the explore/exploit tradeoff). Additionally, a way to balance the two, which requires a schedule, with an exponential function with two hyperparameters for the user to choose. Also provides a regret bound.
- Claim improved results, though this isn’t clear.

**Strengths:**

- Fig. 1 provides a nice illustration of how the algorithm samples promising candidates but does not ultimately return them. The identification error, though not motivated or defined super clearly, seems like an important quantity to address. My understanding is that it represents the effect of noise / distribution shift on optimization solution.
- It’s interesting that minimizing the expected identification error at the next step leads to the difference of EI and KG acquisition functions. It’s also implementationally and theoretically convenient.

**Weaknesses:**

- Doesn’t clearly articulate in the abstract/introduction why or how one would end up in a situation where the algorithm has identified promising regions of the search space but cannot identify/return them. Overall, the writing could use more work to clearly articulate the authors’ arguments and method.
- Distinguishes BO from BAI / R&S by saying that in BO, the candidate set expands as the algorithm explores the search space. This doesn’t make sense; in BO, the search space is usually pre-defined, though different areas may be evaluated at different rounds. Definitions in related work are generally confusing and perhaps inaccurate — posterior best is the point “with the lowest posterior mean”. The related work doesn’t do a compelling job of convincing the reader that their method is needed. The connection to resource allocation strategies is referred to but never explicitly explained or made clear.
- Only looks at two low-dimension synthetic functions as test cases, so hard to tell if this generalizes or is useful in more real-world cases. Further, discretizes each function to have only 100 candidate points, making it further from a realistic case.
- Fig.2 identification error results are not very compelling — It doesn’t really look like NEI / KG have significantly higher identification errors compared to IDEA / identification loss only acquisitions. This calls into question whether better results, if obtained, would be caused by lower identification error.
- Further, Fig.3,4 are not very impressive—IDEA doesn’t really seem to do much better than baselines, and their confidence intervals overlap. I’m not convinced that their method provides a consistent or substantive improvement. The claims made in the text are strong and don’t seem to be reflected in the results.

General feedback for improving paper:
Please format your citations appropriately, particularly making them parenthetical.
Figure 1 is missing a caption for panel b.
Please print your paper out and then rescale your plots based on that — many of them are way too small to distinguish methods or read the legends.
Also, maybe put your plots on a log-scale so it’s easier for the reader to see what’s going on.

Recommend reject — the results don’t seem to constitute a substantive improvement over existing baselines and writing could be improved a lot.

**Questions:**

What is the distinction between the best-arm identification literature and standard multi-arm bandits? Why not just cite these works?
Can you provide p-values to quantify whether the simple regrets (and quantities in other plots) are significantly different between IDEA and KG, and IDEA and NEI?

---

### Official Review · Reviewer_RM1n · 2025-11-01

**Soundness:** 3
**Presentation:** 2
**Contribution:** 1
**Rating:** 2
**Confidence:** 4

**Summary:**

The paper proposes IDEA, an acquisition function for noisy Bayesian Optimization. Due to noisy observations, choosing the solution with the best observed value found can be suboptimal. In particular, IDEA tackles the identification error, which is the difference between the best solution based on the posterior mean and the best solution based on true objective function value (Eq. 16). The authors point out that this identification error can be minimized by minimizing the difference between EI and KG. Hence, the IDEA acquisition function is proposed, which simultaneously (1) maximizes the EI acquisition function and (2) minimize the EI – KG term. A theoretical convergence analysis is provided. Empirically, IDEA is evaluated with EI and KG on two synthetic benchmark problems with different noise settings.

**Strengths:**

- The mathematical derivation of the final rule (EI – KG) is sound.
- There is a convergence analysis to support the result.

**Weaknesses:**

1.	According to the problem formulation, the identification error arises only at the final stage - after completing the BO process - when selecting the final solution from the observed dataset. Therefore, it seems incorrect to use the rule as an acquisition function during the BO routine. During BO, identification error is not yet relevant; hence, the rule should only be applied when determining the final solution. This significantly reduces the contribution of the work.
2.	In the acquisition function in Eq. 2, it is not clear why we only care about explicitly maximizing EI, and not maximizing KG, or both. The final acquisition function could have 3 goals simultaneously: (1) max EI, (2) max KG and (3) min (EI – KG).
3.	The empirical performance does not show any significant improvement. This further strengthens the Weakness 1 that EI-KG rule does not contribute to the BO process, hence the IDEA method seems to be equivalent to EI.
4.	There are only two synthetic benchmark problems. There should be more problems, especially real-world ones.
5.	Many baselines mentioned in the Related Work section are not included in the experiments, including Quan et al., 2013, Liu et al., 2014 and Dai et al., 2023.
6.	There is no ablation study on the effect of weight settings in Eq. 22.
7.	Minor: In line 294, there are two notations of \mu*_n.

**Questions:**

1.	Did the authors consider formulating Eq. 22 as a multi-objective acquisition function, e.g., consisting of the EI objective and the EI – KG objective? Formulating as a multi-objective optimization problem would remove the weight setting requirement.

---

### Note · Program_Chairs · 2026-01-17
**Submission Desk Rejected by Program Chairs**

The following references in this submission do not refer to real documents and/or have major errors in bibliographic information:

 Chun-Hung Chen, Jyh-Jen Lin, Enver Yücesan, and Stephen E. Chick. Simulation selection for constrained optimization. Operations Research, 48(6):836-848, 2000. doi: 10.1287/opre.48.6. 836.1238